# GNSS-TS-NRS: An Open-Source MATLAB-Based GNSS Time Series Noise Reduction Software

**Xiaoxing He [1]**, **Kegen Yu [2]**, **Jean-Philippe Montillet [3]**, **Changliang Xiong [1,4,\*]**, **Tieding Lu [5]**, **Shijian Zhou [5,6]**, **Xiaping Ma [7]**, **Hongchao Cui [8]** and **Feng Ming [9]**

[1] School of Civil Engineering and Architecture, East China Jiao Tong University, Nanchang 330013, China; hexiaoxing@whu.edu.cn

[2] School of Environment Science and Spatial Informatics, China University of Mining and Technology, Xuzhou 221116, China; kgyu@sgg.whu.edu.cn

[3] Physikalisch-Meteorologisches Observatorium Davos/World Radiation Center (PMOD/WRC), CH-7260 Davos, Switzerland; jean-philippe.montillet@pmodwrc.ch

[4] Innovation Academy for Precision Measurement Science and Technology, Chinese Academy of Science (CAS), Wuhan 430077, China

[5] School of Geodesy and Geomatics, East China University of Technology, Nanchang 330013, China; tdlu@whu.edu.cn (T.L.); shjzhou@nchu.edu.cn (S.Z.)

[6] Nanchang Hangkong University, Nanchang 330063, China

[7] School of Geomatics, Xi'an University of Science and Technology, Xi'an 710054, China; xpmaxkd16@xust.edu.cn

[8] Hubei Land Resources Vocational College, Wuhan 430090, China; hccui@ecut.edu.cn

[9] Xi'an Research Institute of Surveying and Mapping, Xi'an 710054, China; mf_pla@hotmail.com

**\*** Correspondence: xcl@ecjtu.edu.cn

**Abstract:** The global navigation satellite system (GNSS) has seen tremendous advances in measurement precision and accuracy, and it allows researchers to perform geodynamics and geophysics studies through the analysis of GNSS time series. Moreover, GNSS time series not only contain geophysical signals, but also unmodeled errors and other nuisance parameters, which affect the performance in the estimation of site coordinates and related parameters. As the number of globally distributed GNSS reference stations increases, GNSS time series analysis software should be developed with more flexible format support, better human–machine interaction, and with powerful noise reduction analysis. To meet this requirement, a new software named GNSS time series noise reduction software (GNSS-TS-NRS) was written in MATLAB and was developed. GNSS-TS-NRS allows users to perform noise reduction analysis and spatial filtering on common mode errors and to visualize GNSS position time series. The functions' related theoretical background of GNSS-TS-NRS were introduced. Firstly, we showed the theoretical background algorithms of the noise reduction analysis (empirical mode decomposition (EMD), ensemble empirical mode decomposition (EEMD)). We also developed three improved algorithms based on EMD for noise reduction, and the results of the test example showed our proposed methods with better effect. Secondly, the spatial filtering model supported five algorithms on a separate common model error: The stacking filter method, weighted stacking filter method, correlation weighted superposition filtering method, distance weighted filtering method, and principal component analysis, as well as with batch processing. Finally, the developed software also enabled other functions, including outlier detection, correlation coefficient calculation, spectrum analysis, and distribution estimation. The main goal of the manuscript is to share the software with the scientific community to introduce new users to the GNSS time series noise reduction and application.

**Keywords:** software; MATLAB; GNSS; time series; EMD; noise reduction

## 1. Introduction

With the rapid development of space observation technology, the global navigation satellite system (GNSS) has become an important tool to observe and model geophysical processes (e.g., tectonic rate, landslide, earthquake displacement map, seasonal variations), and is being widely used in the study of the crustal deformation of the Earth's surface [1–4]. Currently, the globally distributed international GNSS service (IGS) reference stations have accumulated nearly twenty years of coordinate time series, which provide valuable basic data for the study of the geodynamics and global tectonics of Earth's lithosphere and mantle. Tens of thousands of GNSS stations have been installed worldwide to provide continuous position information with millimeter-level accuracy in order to measure their changes in position over time, which is associated with a number of geophysical phenomena such as the plate motion [5,6], the deformation of the crust due to earthquakes (i.e., pre-, co-, and post-seismic offsets [7]), glacial isostatic adjustment [8,9], ocean tide loading [10,11], and atmospheric loading [12].

However, GNSS time series are a sum of stochastic processes and geophysical signals such as the tectonic rate and the seasonal variations [3,13]. GNSS daily position time series also contain unmodeled signals (e.g., satellite orbit errors, small offsets) which affect the precise estimation of the geophysical signals. Therefore, the analysis of the daily position time series should include the implementation of various filtering models, together with making further systematic studies on the source of these global positioning system (GPS) nonlinear variations. These techniques could not only help in the estimation of geophysical signals, but can also lead to the detection of small amplitude signals and to understanding new geodynamical mechanisms.

Routine analysis of GNSS time series can be performed with specific software. There are various high-precision GNSS data post-processing software packages, such as GGMatlab, iGPS, Sigseg, TSAnalyzer, and SARI [14–18]. There are also several specific packages which are used to estimate the linear trend in time series with temporal corelated noise, e.g., CATS, Hector, and Est_noise [19–21]. Several geodetic groups have developed graphical user interfaces (GUIs) and services creation for GNSS time series modeling and visualization [22–24]. However, only a few software packages were designed specifically for GNSS time series noise reduction and analysis. These software packages suffer from several drawbacks such as a user-friendly interface, or not being independent of commercial software. To address these issues, we are here developing an open-source MATLAB-based GNSS time series noise reduction software (GNSS-TS-NRS) written in MATLAB. Note that GNSS-TS-NRS could also act as a TS input of CATS, Hector, and Est_noise, and the main feature of GNSS-TS-NRS is that it provided powerful noise reduction functions.

## 2. Program Language and Installation

GNSS-TS-NRS was developed by using the MATLAB programming language with a graphical user interface [25]. It has powerful functions in scientific computing, graphic visualization, computer simulation, and the simplicity of programming languages. GNSS-TS-NRS mainly consists of source code files, sample files, and instruction manuals. It also supports independent operation with the MATLAB m-files. The main software interface is shown in Figure 1. It can be seen from Figure 1 that GNSS-TS-NRS is mainly composed of 5 modules: Time series noise reduction module, common mode error mitigation module, a time series plot and statistical analysis module, a time series processing tools module, nearby sites and find co-located sites module. Users can start the package by running "GNSS-TS-NRS.m" in MATLAB.

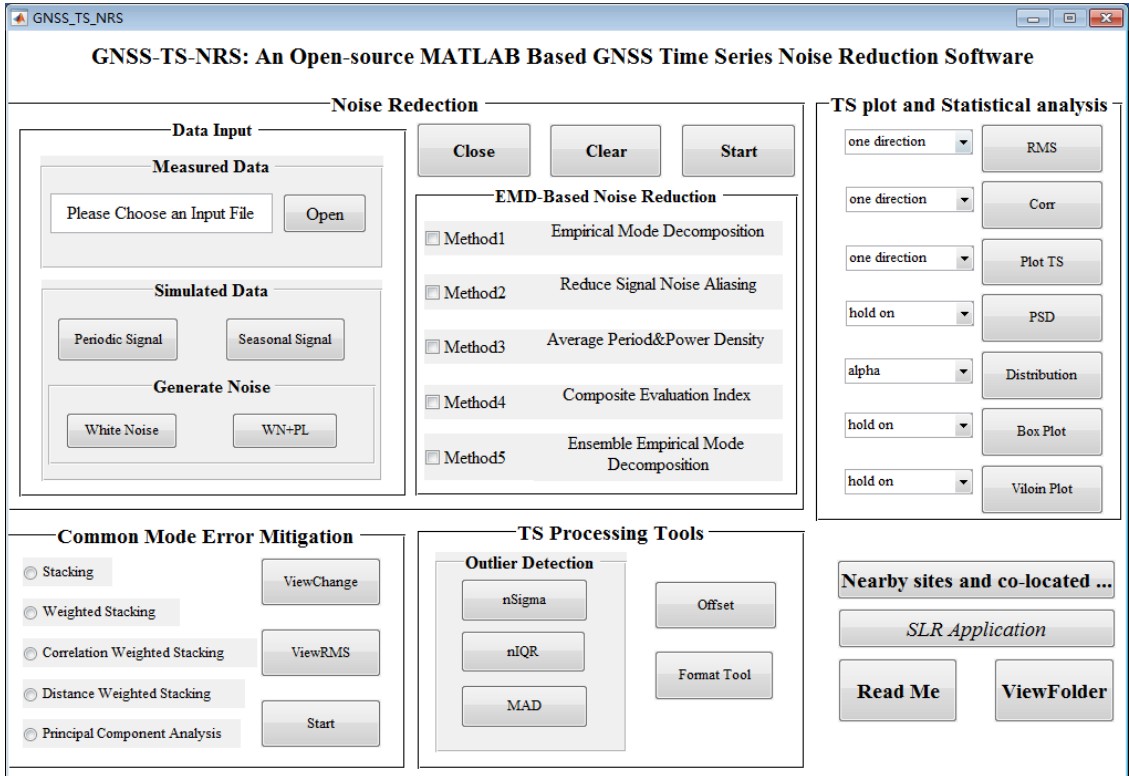

**Figure 1.** Main interface of global navigation satellite system time series noise reduction software (GNSS-TS-NRS).

## 3. Software Features of GNSS-TS-NRS

The models of GNSS-TS-NRS are highly independent, meaning the user can run each model independently. At the same time, there is a certain connection between modules; that is, the output of one module can be used as the input data of another module. The integration and fusion of the modules form a fully functional time series processing and analysis software. The details of the mathematical models and main functions of each module are provided below.

### 3.1. Common Mode Error Mitigation Model

The common mode error mitigation model (spatial filtering analysis model) includes data import, method selection, spatial filtering, graph drawing, and accuracy evaluation. The graph drawing includes three parts: (1) Drawing the time series before and after filtering the common mode error (CME) of each station; (2) accuracy evaluation calculating the root mean square (RMS) of the time series before and after filtering; (3) viewing the rate of RMS change before and after filtering. Five spatial filtering methods are introduced below.

### 3.1.1. Stacking Filtering Method

The basic principle of the stacking filtering method is described as follows. Suppose that there are $S$ GNSS stations in the network: After removing the mean and trend terms of the recorded GNSS time series, the residual time series is obtained. Then the CME is calculated by the following formula [26]:

$$\varepsilon_{SF}(i) = \frac{\sum\limits_{n=1}^{S} \varepsilon_{n,i}}{S} \tag{1}$$

where $i$ is the index of the epoch, $\varepsilon_{n,i}$ is the coordinate residual time series of the $n$-th GNSS station at the $i$-th epoch, and $\varepsilon_{SF}(i)$ is the coordinate residual time series after stacking filtering at the $i$-th epoch. It can be seen that in the stacking filtering method, the CME is equal to the average of the residuals of all the stations in the GNSS network at the current epoch [27].

### 3.1.2. Weighted Stacking Filtering Method

Suppose also that there are $S$ GNSS stations in the network. After removing the mean and trend terms of the recorded GNSS time series, the common mode error at the $i$-th epoch is calculated by the following formula [28]:

$$\varepsilon_{WSF}(i) = \frac{\sum\limits_{n=1}^{S} \frac{\varepsilon_{n,i}}{\sigma_{n,i}^2}}{\sum\limits_{n=1}^{S} \frac{1}{\sigma_{n,i}^2}} \tag{2}$$

where $\sigma_{n,i}$ is the error of the daily position solution of the coordinate residual time series of the $n$-th GNSS station at the $i$-th epoch, and other parameters have the same definitions in Equation (1). When there were less than 3 sites, the calculation of common mode error was not performed.

### 3.1.3. Correlation Weighted Stacking Filtering Method

The correlation coefficient of the residual position time series between GPS stations could well characterize the commonality of CME between the stations. Different from the weighted stacking filtering algorithm, the correlation weighted superimposed filtering separately calculates the individual CME sequence of each station as follows [29]:

$$\varepsilon_{CWSF}(j, i) = \frac{\sum\limits_{n=1}^{S} \frac{\varepsilon_{n,i}}{\sigma_{n,i}^2} \times r_{j,n}}{\sum\limits_{n=1}^{S} \frac{1}{\sigma_{n,i}^2} \times r_{j,n}} \tag{3}$$

where $\varepsilon_{CWSF}(j, i)$ represents the CME value of station $j$ at epoch $i$, and $r_{j,n}$ is the correlation coefficient between the coordinate residual sequences of station $j$ and station $n$.

### 3.1.4. Distance Weighted Filtering Method

It is a fact that that the distance between GNSS stations affects the correlation between the coordinates of the stations, and also has a certain impact on the regional CME size. The distance weighted filtering method uses the center distance of all stations as the weight to calculate the CME of a give station $n$ ($n = 1, 2 \ldots S$) as follows:

$$l_n = (x_n - \overline{x})^2 + (y_n - \overline{y})^2 + (z_n - \overline{z})^2 \tag{4}$$

$$\varepsilon_{DWF}(i) = \frac{\sum\limits_{n=1}^{S} \frac{\varepsilon_{n,i}}{l_n}}{\sum\limits_{n=1}^{S} \frac{1}{l_n}} \tag{5}$$

where $l_n$ is the distance from the station $n$ to the center of the GPS network stations, and $(x, y, z)$ is the mean position of the stations.

### 3.1.5. Principal Component Analysis

Principal component analysis (PCA) is a standard mathematical tool that transforms a number of different, but possibly correlated, variables into a smaller number of uncorrelated variables called

principal components. Dong et al. proposed a PCA spatial filtering method, also called eigenvector analysis, which is a commonly used data analysis method to separate common mode errors [30]. The core idea is to transform the original data into a set of linearly independent representations of each dimension through linear transformation; that is, to orthogonally decompose the data into mutually orthogonal vector spaces, which can be used to extract the main components (principal components) of the data features [31,32].

*3.2. Noise Reduction Analysis Model*

Signal noise reduction includes four steps: Data import, noise reduction processing, graph drawing, and precision analysis. The data import part imports external data or generates simulation data. The noise reduction analysis part uses the correlation coefficient, root mean square error, and signal-to-noise ratio as evaluation indicators to quantitatively evaluate the noise reduction effect. The correlation coefficient reflects the similarity between the denoising time series (TS) and the original TS. The closer the correlation coefficient value is to 1, the better the fitting effect; that is, the better the denoising effect. Secondly, the root mean square error reflects the degree of deviation between the denoising signal and the original signal, and the smaller the value, the better the denoising effect. Thirdly, the signal-to-noise ratio mainly reflects the proportion of the noise signal in the overall signal. The larger the value, the better the denoising effect. Last, the graph drawing function can draw the correlation coefficient graph of the *IMF* (intrinsic mode function) and the original time series, each *IMF* component graph, and the signal sequence comparison graph before and after noise reduction. In the following section, we focus on the five noise reduction algorithms built into the software.

3.2.1. Empirical Mode Decomposition (Method 1)

The empirical mode decomposition (EMD) method was first proposed by Huang [33]. It is an adaptive signal processing method for non-stationary nonlinear signals. The steps of EMD decomposition of signal $X(t)$ are:

Step 1: Find all the maximum and minimum points of the original time series $X(t)$. Calculate the average of the upper and lower envelopes $d_1$. Subtract $d_1$ from the original time series, then attain a new time series $c_1(t)$;

Step 2: Repeat step 1 until the *IMF* threshold condition is met, to attain the first *IMF* component;

Step 3: Subtract the first *IMF* component from the original data sequence $X(t)$ to form a new data sequence $X_2(t)$, then repeat steps 1 and 2 until the $m-$th *IMF* component is obtained. Thus, the original data sequence can be expressed as:

$$x(t) = \sum_{k=1}^{m} IMF_k + r(t) \tag{6}$$

where $r(t)$ is the residual term, $k$ is the *IMF* serial number, and $t$ means the epoch.

In actual signal decomposition, the *IMF* component is difficult to strictly meet the condition that the upper and lower envelopes composed of local extrema have zero mean value. The threshold expression for each $j$-th to stop filtering is given as follows:

$$SD = \sum_{t=0}^{N-1} \left[ \frac{(c_k(t) - c_{k-1}(t))^2}{c_k(t)^2} \right] \tag{7}$$

*SD* is the threshold for each *IMF* to stop sifting, usually 0.2~0.3. $c_k(t)$, $c_{k-1}(t)$ are the two adjacent data sequences of the *k*-th *IMF* sifting process.

After the original data sequence is decomposed by EMD, it is necessary to determine the boundary *IMF* function between the noise and the real signal. The boundary *IMF* is selected by the correlation

coefficient criterion. That is, the *IMF* corresponding to the minimum value of the correlation coefficient $\rho_k$ for the first time is the boundary *IMF* function, which is calculated as follows:

$$\rho_k = \frac{\sum\limits_{t=0}^{N-1} IMF_k(t)x(t)}{\left[\sum\limits_{t=0}^{N-1} IMF_k^2(t) \sum\limits_{t=0}^{N-1} x^2(t)\right]^{\frac{1}{2}}} \tag{8}$$

The EMD method incorporates the boundary *IMF* into the noise part, and the low-frequency *IMF* components and residual terms after the demarcation are reconstructed to obtain a noise-reduced signal, which can be expressed as:

$$\hat{x}(t) = \sum_{k=K+1}^{m} IMF_k + r(t) \tag{9}$$

where $\hat{x}(t)$ is the signal after noise reduction and $K$ is number of the boundary *IMF* value. For $IMF_k$, if $K < k \leq m$, we take the $IMF_k$ as the signal component, and if $k \leq K$, then we consider the $IMF_k$ as noise.

### 3.2.2. Signal Noise Aliasing Reduction (Method 2)

The traditional EMD method determines the boundary *IMF* according to the correlation coefficient rule after obtaining *IMFs*, then deleting the boundary *IMF* and the previous judgment as high-frequency noise components, to achieve the separation of signal and noise. However, due to the problem of signal-to-noise aliasing, high-frequency *IMF* may still contain some real signal. Based on this, we proposed a method of using multiple EMD to reduce signal noise. The detailed process is as follows [34]:

Step 1: Initialization: Download raw data $x_i$, $i = 1$;
Step 2: EMD decomposition to obtain $M_i$ *IMF*, and trend items $r_i(t)$;
Step 3: Calculate the correlation coefficient; the boundary *IMF* is $IMF_{K_i}$;
Step 4: Low-frequency *IMF* reconstruction $t_i = \sum_{m_i=K_i+1}^{M_i} IMF_{m_i} + r_i(t)$;
Step 5: Eliminate the first high frequency *IMF*;
Step 6: $i = i + 1$;
Step 7: High-frequency *IMF* reconstruction: $x_i = \sum_{m_i=2}^{K_i} IMF_{m_i}$;
Return to step 2.

The cyclic decomposition stops when the dividing *IMF* value is $k_i = 2$ or the correlation coefficient is monotonous. Then, all low-frequency *IMF* components and trend items are reconstructed to obtain the data sequence after noise reduction, which can be expressed as:

$$\hat{x}_1 = \sum_{m_1=K_1+1}^{M_1} IMF_{1m_1} + \sum_{m_2=K_2+1}^{M_2} IMF_{2m_2} + \cdots + \sum_{m_i=K_i+1}^{M_i} IMF_{im_i} + \sum_{j=1}^{i} r_j(t) \tag{10}$$

In the formula, $\hat{x}_1$ is the time series after noise reduction, $K_i$ represents the order of noise coexistence of the *i*-th original data sequence, $M_i$ represents the total number of *IMF* components of the *i*-th original data sequence, $IMF_{im_i}$ represents the *IMF* component of the *i*-th original data sequence, and $r_j(t)$ represents the residual term of the *j*-th EMD decomposition.

### 3.2.3. Average Period and Power Density (Method 3)

Considering the complexity of using the correlation coefficient criterion to determine the demarcation *IMF*, and the issue of selecting the $K$ value of the delimited *IMF* function, we proposed using the product of the average period and energy density as an indicator, and an algorithm for

automatically determining the boundary *IMF* was developed, which could directly determine the *K* value of the boundary *IMF* function [35].

The average period $\overline{T}_k$ of the *k*-th *IMF* is given by:

$$\overline{T}_k = \frac{N \times 2}{nem_k} \tag{11}$$

and the energy density is calculated by:

$$E_k = \frac{1}{N} \sum_{t=0}^{N-1} [IMF_k(t)]^2 \tag{12}$$

Thus, the product of the average period and the energy density is:

$$ET_k = E_k \times \overline{T}_k \tag{13}$$

Here, $nem_k$ is the total number of extreme points of the *k*-th *IMF*, and *N* is the data length of each mode.

The threshold for determining the boundary between signal and noise is defined as:

$$R_{k-1} = abs\left( \frac{ET_k}{\frac{1}{k-1} \sum_{i=1}^{k-1} ET_i} \right) \tag{14}$$

where abs(*x*) denotes the absolute value of *x*, *k* ≥ 2, when $R_{k-1} \geq C$ (here *C* is 2). Then we determined *k* as the boundary point, reconstructing the first *k* − 1 noise-dominated modal components, and the data sequence was subtracted from the reconstructed noise to obtain the noise-reduced signal.

### 3.2.4. Composite Evaluation Index (Method 4)

To reduce the complexity of determining the boundary *IMF*, based on the correlation coefficient criterion, and dealing with the inaccuracy of using the single index to determine the boundary *IMF*, a composite evaluation index (*T*) was adopted. This index comprehensively considered the two index values of curve smoothness (*r*) and root mean square error (RMSE), and an improved EMD noise reduction method is provided to directly determine the *K* value of the delimited *IMF* function.

For formula clarity, the residual term was regarded as the last *IMF* component.

The formula for calculating the root mean square error is given by [36]:

$$\text{RMSE} = \sqrt{\frac{1}{N} \sum_{t=0}^{N-1} (IMF_k(t) - x(t))^2} \tag{15}$$

and the formula for calculating smoothness is defined as:

$$r = \frac{\sum_{t=0}^{N-2} (IMF_k(t+1) - IMF_k(t))^2}{\sum_{t=0}^{N-2} (x(t+1) - x(t))^2} \tag{16}$$

Here, $IMF_k(t)$ represents the *k*-th *IMF* component, the value of *k* is 1, 2, …, *m* + 1; *x*(*t*) represents the noisy original data sequence.

The two indicators of RMSE and *r* are normalized according to:

$$\text{PRMSE} = \frac{RMSE}{\sum (RMSE)} \tag{17}$$

$$\text{Pr} = \frac{r}{\sum (r)} \tag{18}$$

The coefficient of variation in the weighting method is applied to weigh the two normalized indicators, and the weights are defined as follows:

$$CV_{P_{RMSE}} = \frac{\sigma_{PRMSE}}{\mu_{PRMSE}} \tag{19}$$

$$CV_{Pr} = \frac{\sigma_{Pr}}{\mu_{Pr}} \tag{20}$$

$$W_{PRMSE} = \frac{CV_{PRMSE}}{CV_{PRMSE} + CV_{Pr}} \tag{21}$$

$$W_{Pr} = \frac{CV_{Pr}}{CV_{PRMSE} + CV_{Pr}} \tag{22}$$

where $\sigma$ and $\mu$ represent the standard deviations and mean values of the data sequence $\{\text{PRMSE}_k\}$ and $\{Pr_k\}$, respectively; $CV$ represents the coefficient of variation; and $W$ is the weighted value based on the coefficient of variation.

Thus, the composite evaluation index $T$ is described by:

$$T = W_{PRMSE} \times PRMSE + W_{Pr} \times \text{Pr} \tag{23}$$

The threshold for determining the boundary between signal and noise is defined by:

$$R_{k-1} = \left| \frac{T_{k-1}}{T_k} \right| \tag{24}$$

where $k \geq 2$, and when the threshold first satisfies $1 \leq R_{k-1} \leq 3$, then $k - 1$ is determined as the cutoff point; that is, the $K$ value of the cutoff *IMF* function is $k - 1$, and the previous $K$ *IMF* components ($IMF_j$, $j \leq K$) are all regarded as noise, and the *IMF* components ($IMF_j$, $j > K$) are regarded as a signal. The *IMF* components dominated by the signal are reconstructed to obtain a sequence after noise reduction.

### 3.2.5. Ensemble Empirical Mode Decomposition (Method 5)

To reduce the influence of mode aliasing, Wu and Huang also developed a noise-assisted EMD method called ensemble empirical mode decomposition (EEMD) [37]. First, this method adds white noise of finite amplitude to the signal:

$$x_i(t) = x(t) + w_i(t) \tag{25}$$

where $x(t)$ represents the observation, $x_i(t)$ represents the *i*-th observation perturbed by white noise, and $w_i(t)$ represents the white noise that is added to the *i*-th observation. The magnitude of the added white noise is decided on the ratio of the standard deviation of the added noise and that of $x(t)$, which is the ratio of the standard deviation of the added noise and that of $x(t)$ and is usually about 0.1 or 0.2. On the other hand, the number of the ensemble size is several hundreds. Each decomposition will generate a set of *IMFs* from high frequency to low frequency. Since white noise is a normal random sequence with zero mean, the influence of adding white noise will be eliminated by averaging. The corresponding *IMFs* are averaged to obtain the final decomposed *IMFs* of EEMD [38]. Since white

noise is a normal random variable with zero mean, after the ensemble size times average, the influence of adding white noise is eliminated, and the decomposed *IMFs* all come from the original signal itself.

### 3.3. Time Series Processing Tools

#### 3.3.1. GNSS Time Series Format Convert

There are many GNSS time series data formats, so format conversion tools are very necessary. This module supports the batch conversion of two *.neu format files issued by the crustal movement observation network of China (CMONOC) and Scripps orbit and permanent array center (SOPAC) into *.mom format files, which can be used by other modules in the software.

#### 3.3.2. Offset Correction and Analysis

The estimation of geophysical signals such as uplift rates and seasonal signals are affected by discontinuities in the GNSS time series that can be caused by equipment changes at individual stations as well as earthquakes producing spatially coherent offsets [19,26,39]. In the current version of GNSS-TS-NRS, offset detection is not implemented, but it supports offset correction with known parameters; e.g., the offset parameter delivered by SOPAC or site log files delivered by IGS data centers. We can also use Hector to make the offset estimation and then make the offset correction based on the estimated offsets [20]. Corresponding to the offset correction function, we added an artificial offset function, which could add offsets in time series in batches, and could be used to explore the relation or effect of offset on geophysical signals. Figure 2 is an example of the offset, which inserted four offsets with an amplitude of 18 mm in four epochs.

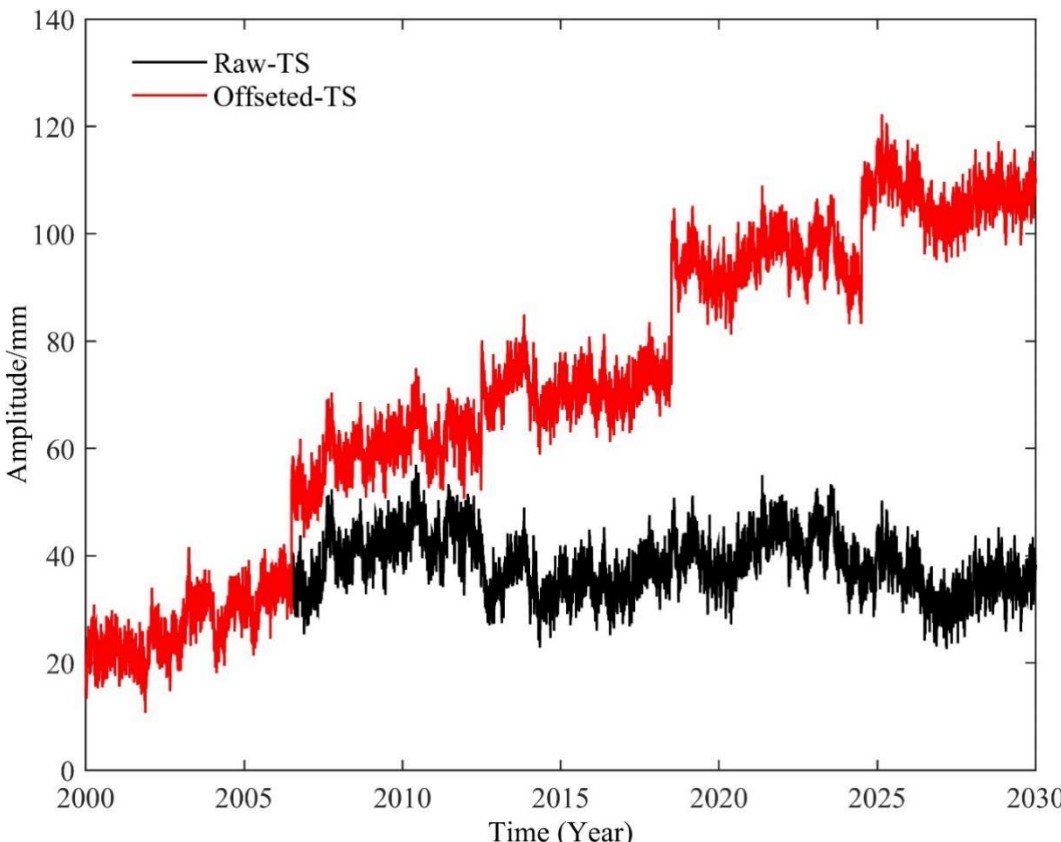

**Figure 2.** The result of TEST_1 after inserting an 18 mm (offset) step function in 4 epochs.

### 3.3.3. Outlier Detection Function

This module contains 4 algorithms, i.e., 3 Sigma, 5 Sigma, 3 interquartile range (IQR), and median absolute deviation (MAD) [40], which can read time series files in batches, detect gross errors according to the selected algorithm, remove them after determining them as gross errors, and save the new file to the corresponding folder. The gross error points on the time series graph are marked, the gross error rate is calculated, and a gross error elimination report is generated.

### *3.4. Time Series Plot and Statistical Analysis*

### 3.4.1. Root Mean Square Calculation

We read the station time series files in batches, calculated the root mean square (RMS) of the time series, plotted the calculation results as shown in Figure 3, and saved the results to files. Figure 3 shows the RMS results of 10 CMONOC stations from 1999 to 2019, which can reflect the stability of the time series.

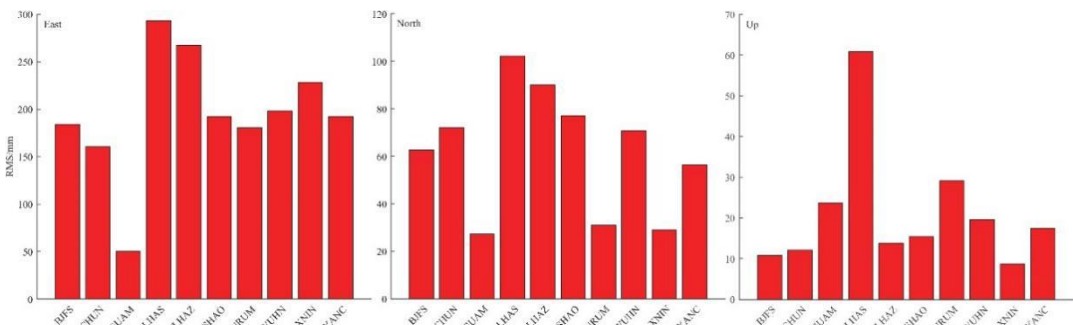

**Figure 3.** The root mean square (RMS) of ten sites in three coordination directions.

### 3.4.2. Correlation Coefficient Calculation

The correlation coefficient is an indicator that reflects the correlation between the time series of different sites. The software is designed with a module to calculate this indicator. The specific implementation of the calculation is given as follows:

$$Corr(i,j) = \frac{\sum\limits_{t=1}^{N} x_i(t) x_j(t)}{\sqrt{\left(\sum\limits_{t=1}^{N} x_i^2(t)\right)\left(\sum\limits_{t=1}^{N} x_j^2(t)\right)}}, \quad i,j = 1,2\ldots S \tag{26}$$

where $Corr(i,j)$ is the correlation coefficient of the *i*-th and *j*-th site time series, $S$ is the number of stations in the input data, $N$ is the number of epochs in the time series, and $x_i(t)$ is the time series of the *i*-th station. This module can read time series files in batches, calculate correlation coefficients in the same direction between different stations, generate a correlation coefficient matrix, and automatically save the results.

### 3.4.3. Plot GNSS Time Series

There are two data import methods, which import one-direction and three-direction time series, respectively, and can quickly achieve the visualization of time series graphics. Figure 4 shows an example of visualization of a three-direction time series (BJFS, Beijing Fangshan GPS station) from 1999 to 2019, and it can be seen from Figure 4 that BJFS has a long-term trend moving to the southeast, and the vertical component shows seasonal variation.

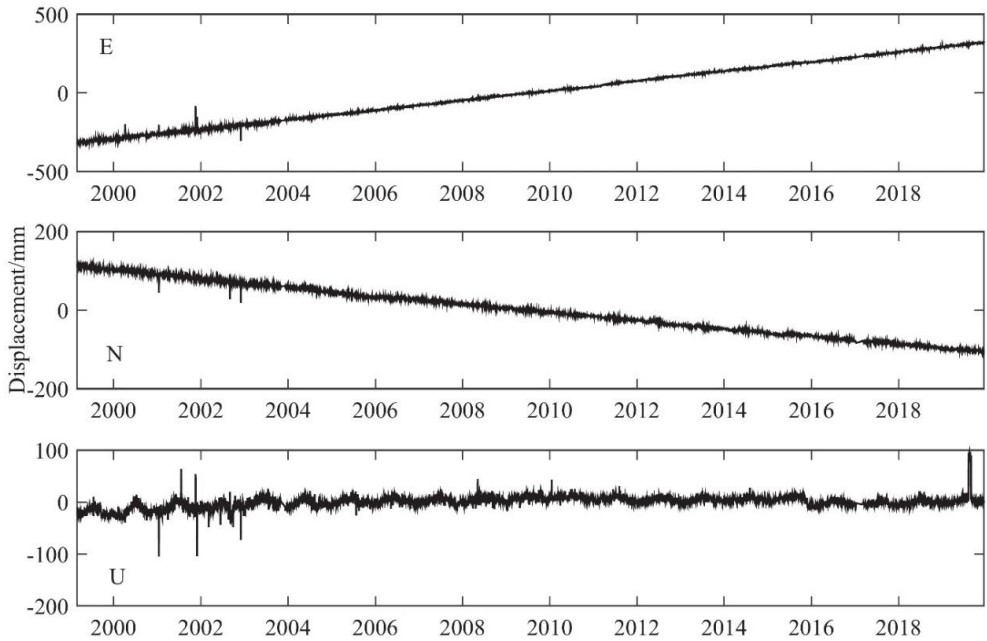

**Figure 4.** The times series visualization of BJFS GPS station.

### 3.4.4. Box-Whisker Plot and Violin Plot Statistics

A box plot does not require that the data to obey a specific distribution in advance, or any other restrictive requirements on the data, and it is not affected by outliers. The box plot can be used to intuitively read the maximum, minimum, median, upper quartile, lower quartile, outlier, and average value of a set of data. There is a corresponding module in GNSS-TS-NRS, which can quickly realize batch drawing of data (e.g., 5 stations) box diagrams as shown in Figure 5 [41]. The box diagram can display the data dispersion and intuitively display the mean, maximum, minimum, median, upper, and lower quartiles of the time series. It is mainly used to reflect the characteristics of the original data distribution, and it can also compare multiple sets of data distribution characteristics.

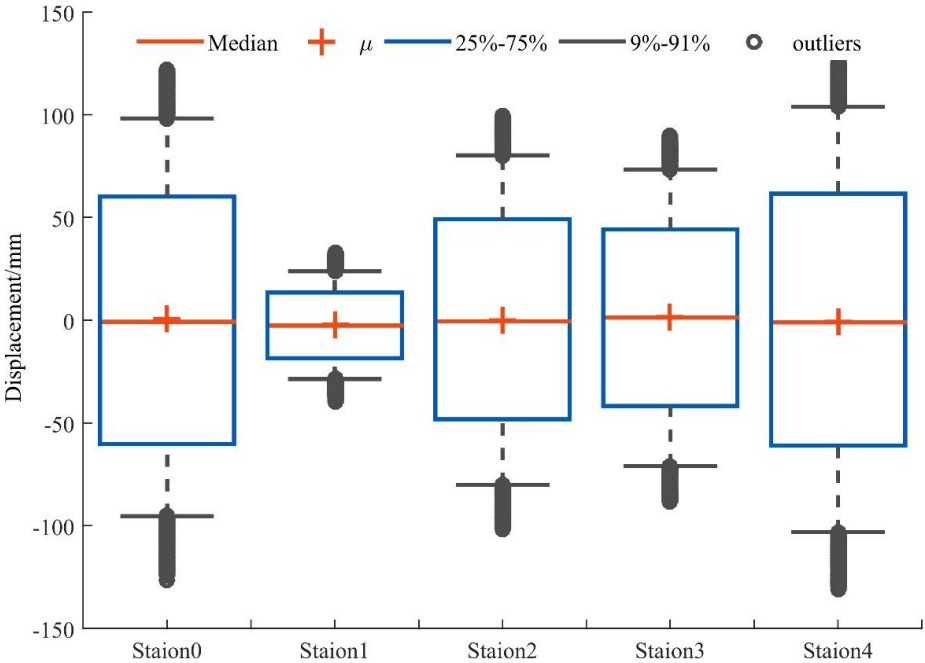

**Figure 5.** The box plot of station 0~4 in the east direction.

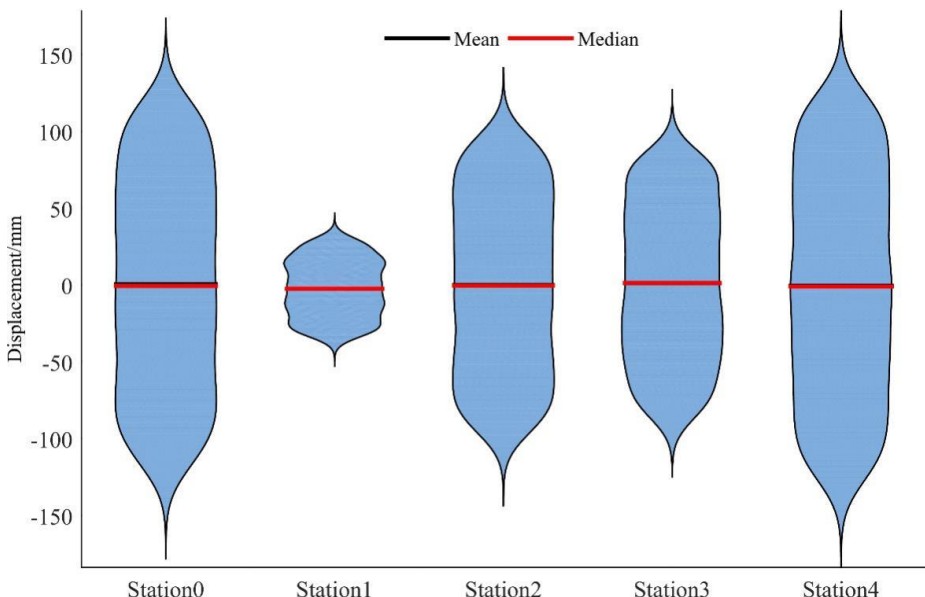

**Figure 6.** The violin plot of station 0~4 in the east direction.

On the other hand, a violin plot is considered as a combination of the box plot with a kernel density plot, and is mainly used to show the distribution shape of data. It is similar to the box plot, but is better displayed at the density level. The violin chart is especially suitable when the amount of data is very large, and it is not convenient to display one by one. The drawing using the same data with box plot is shown in Figure 6 [42]. The violin chart is usually used to show the distribution status and probability density of multiple sets of data. This kind of chart combines the characteristics of a box plot and a density plot, and is mainly used to show the distribution shape of the data. It is similar to the box plot, but is better displayed at the density level. The violin chart is especially suitable when the amount of data is very large, and it is not convenient to display one by one.

### 3.4.5. Power Spectral Density Analysis

Power spectral density (PSD) is a physical quantity that characterizes the relationship between the power energy of the signal and the frequency. PSD is usually normalized according to the frequency resolution, and is often used to study random vibration signals. There is a module in GNSS-TS-NRS that can quickly draw PSD images of GNSS time series for data analysis. Figure 6 shows an example of the PSD plots of six coordinate time series using fast Fourier transform (FFT), which was employed to evaluate the dominant periods of these signals in the frequency domain, especially for periodic signal detection [43,44].

### 3.4.6. Distribution Estimation

It is generally believed that the error of GNSS time series conforms to the normal distribution, but current research shows that the Alpha-stable distribution may be a more reasonable error model [45,46]. Therefore, in GNSS-TS-NRS, there is a module for estimating the distribution of GNSS residual time series, including normal distribution and alpha-stable distribution for users to choose. The module can calculate relevant parameters and draw their cumulative distribution function (CDF) and probability density function (PDF); correlation coefficients between fitted data and original data can be calculated and related graphics can be drawn. Figure 7 shows an example of the PDF and CDF of one-direction time series at a station, generated by GNSS-TS-NRS [47]. From Figure 8, we can see that the Anderson–Darling test checks whether the sample data comes from a specific distribution of ACSO GPS station (which located in Ohio State of United States): h = 0 means that the null hypothesis with 95% confidence is passed, and h = 1, means that the null hypothesis is not true. p is the probability

of observing a test statistic as extreme as, or more extreme than, the observed value under the null hypothesis [48]. Both normal distribution and alpha-stable distribution can be used to model this time series well, although the correlation coefficient between the latter and actual data is slightly larger.

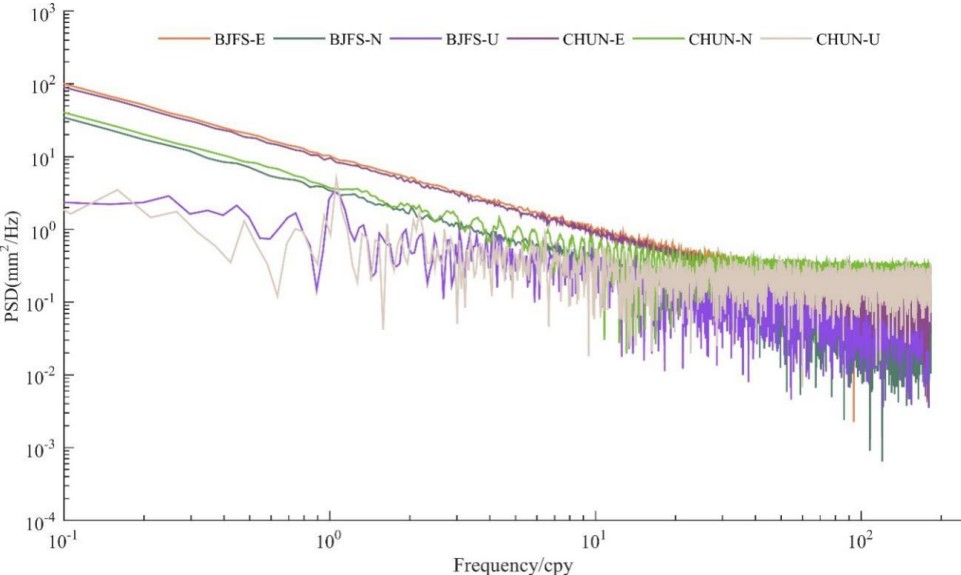

**Figure 7.** Power spectral density (PSD) of three-direction time series at BJFS and CHUN (ChangChun GPS Station) from 1999 to 2019.

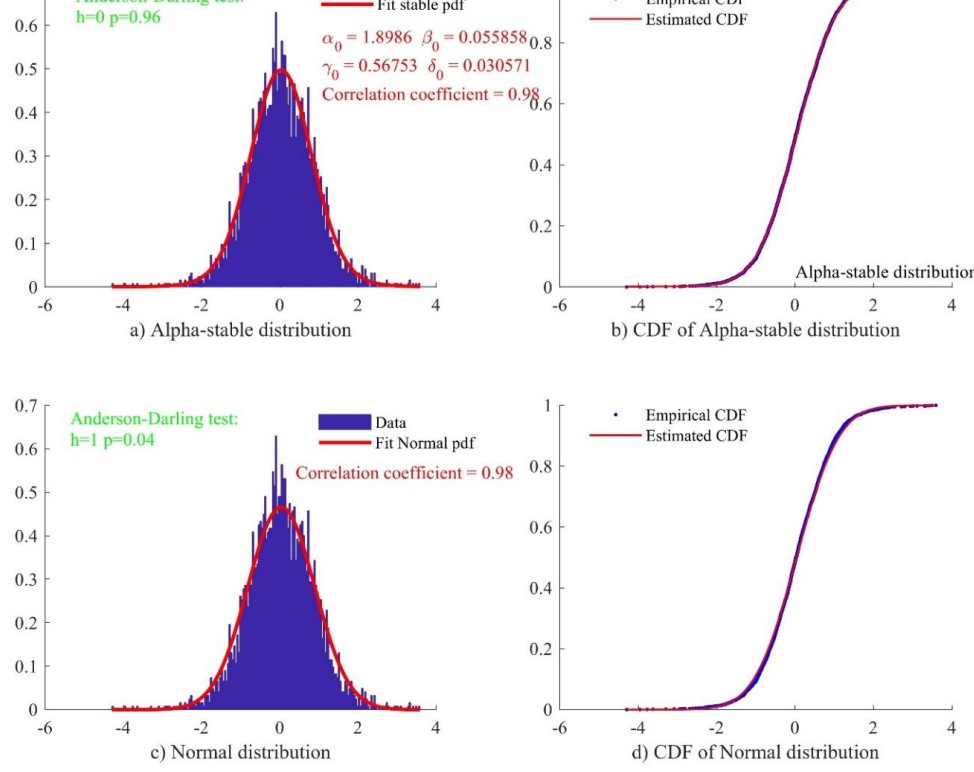

**Figure 8.** Probability density functions and cumulative distribution function of north-direction time series at ACSO GPS station (N coordinate). (**a**) Residual time series with alpha-stable distribution; (**b**) cumulative density function of residual time series and alpha-stable distribution (correlation; alpha = 0.98); (**c**) residual time series with normal distribution; (**d**) cumulative density function of residual time series and normal distribution (correlation; norm = 0.98).

### 3.5. Nearby Sites and Finding Co-Located Sites

This module includes two functions: Searching for nearby sites and judgment of co-located sites [49]. The nearby site search module involves three procedures: (1) Database preparation: Users can create a site database which includes the latitude, longitude, and XYZ coordinates of monitoring network sites such as IGS, CMONOC, and so on; (2) parameter setting: Users can find nearby sites in the database by entering the site name or latitude and longitude, and the limit distance constraint can be set at the same time; (3) operation: Users can click the button "Get nearby sites" to obtain sites that meet the requirements. By use of the station overlay judgment module, users can build the GNSS station and tide gauge (TG) station database, respectively, and enter the limit distance (e.g., 15 [50], 20 km [51]). The spatial distance limitation ensures as possible that the GNSS and the TG sites are close enough, and therefore the GNSS vertical motion can directly contribute to the TG-derived trend (velocity, etc.). Then, the software judges whether the station is overlaid and output it based on the limit distance. Figure 9 shows the software interface for searching for nearby sites and finding co-located sites.

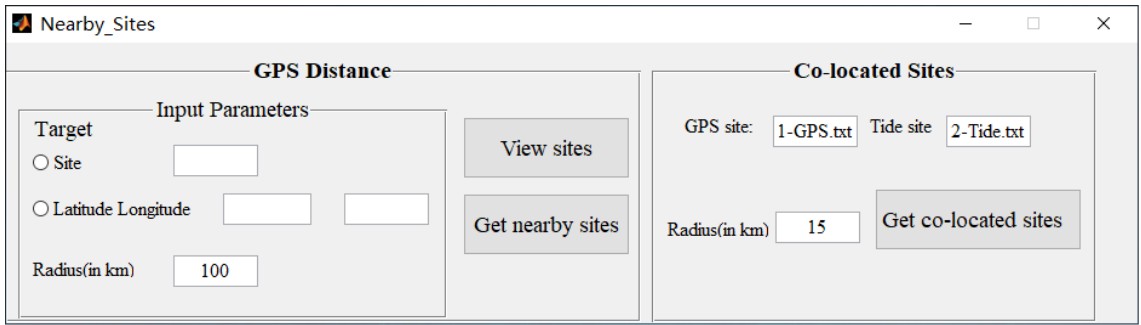

**Figure 9.** Searching for nearby sites and finding co-located sites.

## 4. Noise Reduction Test and Result

In order to verify the reliability of the software and the credibility of the algorithm, specific experiments were carried out on this software platform. The experiments were divided into two parts: The simulation experiment and the measured data experiment to comprehensively test the software. Among them, 3 sets of simulation data, including different types of simulation data and different types of noise, were used to verify the universality of the noise reduction method. The measured data verifies the practicability of GNSS-TS-NRS.

### 4.1. Simulated Test with GNSS-TS-NRS

GNSS coordinate time series are generally composed of three parts: Seasonal items, trend, and noise. In the simulation data I, the site position, trend term, and step offset were first eliminated, and the three constant amplitude periodic terms and noise were mainly considered. Setting the sampling frequency to 1 HZ and setting the number of sampling points to 1024, and then adding the Gaussian white noise so that the signal-to-noise ratio was 4 dB, the simulation data was generated as follows:

$$
\begin{cases}
y_1 = 4\sin(2\pi t/800)\sin(2\pi t/250) \\
y_2 = 2\sin(2\pi t/600) \\
y_3 = \sin(2\pi t/50) \\
\varepsilon = Noise \\
x = y_1 + y_2 + y_3 + \varepsilon
\end{cases}
\tag{27}
$$

where $y_1$, $y_2$, and $y_3$ are the periodic components, and $\varepsilon$ is the added noise. After inputting the above-mentioned parameters in the GNSS-TS-NRS signal input module, the corresponding simulation signal sequence was established by the software, and the graph of each signal component was drawn during operation, as shown in Figure 10.

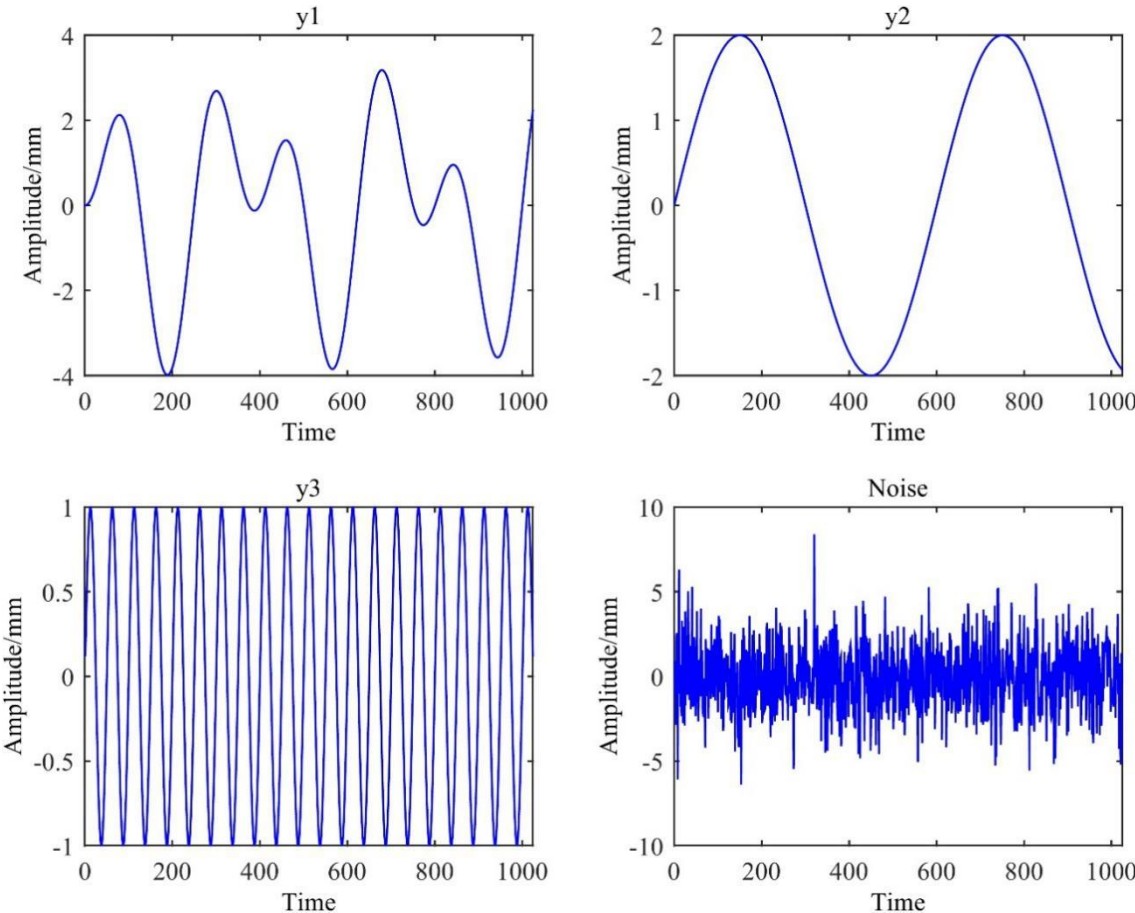

**Figure 10.** Four signal components of simulation data I.

Since the continuous GNSS coordinate sequence contains the amplitude time-varying seasonal signal, the remaining two simulation data were added with the amplitude change factor on the premise of excluding the station position, trend item, and step offset. The time-varying seasonal signal data of the coordinate sequence of 10 years was simulated by the following formula:

$$
\begin{aligned}
y(t_i) = &\, a\sin(2\pi t_i) + b\cos(2\pi t_i) + c(t_i)\sin(2\pi t_i) + c(t_i)\cos(2\pi t_i) + \\
&\, d\sin(4\pi t_i) + e\cos(4\pi t_i) + c(t_i)\sin(4\pi t_i) + c(t_i)\cos(4\pi t_i) + \varepsilon(t_i)
\end{aligned}
\tag{28}
$$

where $y(t_i)$ is the simulated time-varying seasonal signal, $a, b, d, e$ are constants, $t_i$ is the GNSS day of year, $c(t_i) = 2e^{0.3\sin(t_i)}$ is the amplitude change factor, and $\varepsilon(t_i)$ is the noise.

In order to verify the processing effect of different noise types and the function of this software, in simulation data II and III, the values of $a, b, d, e$ were 6, 7, 8, and 9 mm, respectively. The noise added in the simulation data II was Gaussian white noise with a signal-to-noise ratio of 4 dB, and the noise added in the simulation data III was a combination of power law and white noise (PL+WN). The amplitude of white noise was 5 mm, the amplitude of colored noise was 0.02 mm, and the spectral index of power law noise was −1.2. In Figure 11, the two components of the simulation data are drawn separately.

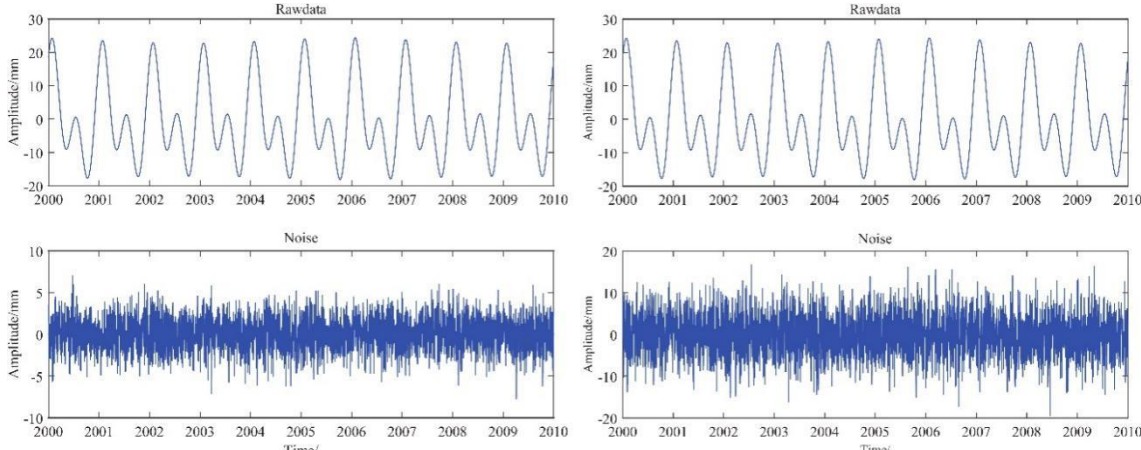

**Figure 11.** Simulation data II (**left**) and III (**right**) are the signal sequences on the top; the left and right are the same. The bottom is the respective noise, the left is white noise of 4 dB, and the right is the power law- and white noise (PL+WN)-type noise (white noise amplitude is 5 mm, colored noise amplitude 0.02 mm, the spectral index of power law noise is −1.2).

In GNSS-TS-NRS, the 3 sets of simulation data were respectively processed for noise reduction by the four methods. Since the true value of the simulation data was known, the noise reduction effect could be quantitatively reflected by calculating the correlation coefficient between the signal sequence after noise reduction and the true value. The calculation results were shown in Table 1.

**Table 1.** The analysis results of the 4 kinds of noise reduction methods after the noise reduction of different data. The indexes are the correlation coefficient, the root mean square error, and the signal-to-noise ratio.

| Method | I | | | II | | | III | | |
|---|---|---|---|---|---|---|---|---|---|
| | $\rho_k$ | RMSE | SNR | $\rho_k$ | RMSE | SNR | $\rho_k$ | RMSE | SNR |
| 1 | 0.9303 | 1.0033 | 7.4029 | 0.9983 | 0.6978 | 299.4933 | 0.9900 | 1.7208 | 50.0330 |
| 2 | 0.9726 | 0.6156 | 18.4766 | 0.9984 | 0.6837 | 311.2626 | 0.9900 | 1.7155 | 50.2418 |
| 3 | 0.9696 | 0.6613 | 16.6838 | 0.9991 | 0.5135 | 552.3157 | 0.9949 | 1.2186 | 98.1886 |
| 4 | 0.9488 | 0.8582 | 10.0070 | 0.9983 | 0.6978 | 299.4933 | 0.9949 | 1.2186 | 98.1886 |

Since the three new methods were based on EMD, we compared the noise reduction effect between EMD and Methods 2–4. As seen in the first and second rows in Table 1, Method 2 was used to reduce the noise of 3 groups of simulation signal sequences. The closer the $\rho_k$ is to 1, the higher the similarity of the time series, the better the fitting effect, and the better the denoising effect. The RMSE reflects the degree of deviation between the noise reduction signal and the real signal, and the smaller the value, the better the noise reduction effect. The signal-to-noise ratio (SNR) mainly reflects the proportion of the noise signal in the overall signal, and the larger the value, the better the denoising effect. In the test of simulation data I, the $\rho_k$ increased by 0.0423, the RMSE decreased by 0.3877, and the SNR increased by 11.0736; in the test of simulation data II, the indicators improved slightly, with the $\rho_k$ increasing by 0.0001, the RMSE being reduced by 0.0140, and the SNR increasing by 11.7693. The test of simulation data III achieved basically similar results. In summary, the simulation data I (the blue line) improved the most, and the signal after Method 2 noise reduction was very smooth, as shown in Figure 12. Therefore, based on the above analysis, we have reason to believe that Method 2 is more helpful for signal noise reduction than the traditional EMD method.

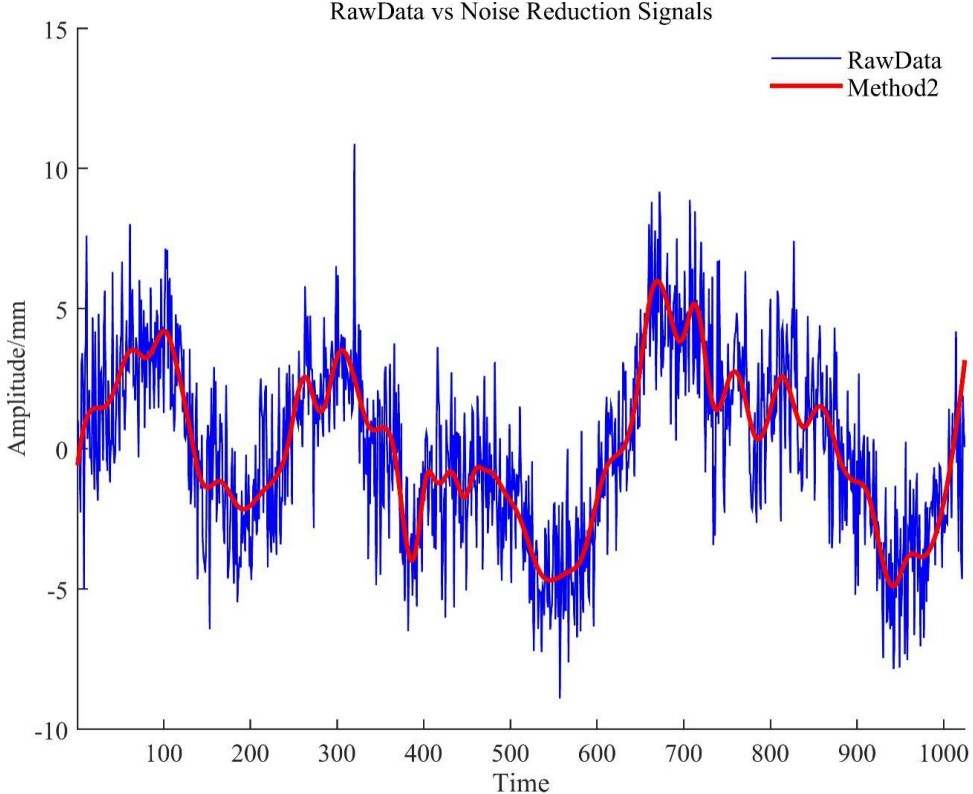

**Figure 12.** Result of noise reduction with method of simulation data I.

On the other hand, Methods 3 and 4 in the software were proposed to solve the problem of inaccurate demarcation by the correlation coefficient method. A simulation experiment was carried out with the same data, and the accuracy evaluation results of the experiment are shown in the third and fourth rows of Table 1. Table 2 shows the boundary *IMF* values determined by the three methods. When Method 3 was applied to simulated data I, the same results as Method 1 were obtained, indicating that Method 3 has a certain degree of reliability. In the test of simulated data II and III, the boundary *IMF* values obtained by Method 3 were different than Method 1. Therefore, we focused on the indicators in Table 1 to determine which method is better. It can be seen that the time series obtained by Method 3 achieved a higher correlation coefficient and SNR and lower RMSE, indicating that this method performs better than the correlation coefficient method of traditional EMD.

**Table 2.** The boundary *IMF* of Methods 1, 3, and 4 in three simulation data.

| Method | Boundary *IMF* Value | | |
|:---:|:---:|:---:|:---:|
| | I | II | III |
| 1 | 5 | 3 | 3 |
| 3 | 3 | 4 | 4 |
| 4 | 4 | 3 | 4 |

The results obtained by Method 4 are similar to the traditional EMD method but also have different results. The difference from the results obtained in Method 3 is that in the experiment with simulation data II, Method 4 failed to obtain the same accurate results as Method 3, but was similar to the EMD method. However, in the experiments with simulated data I and III, the two methods achieved similar results. Therefore, it can be considered that Method 4 outperforms the traditional EMD method. In order to directly compare the noise reduction effect, we provide the noise reduction comparison result of simulation data I using different methods in Figure 13.

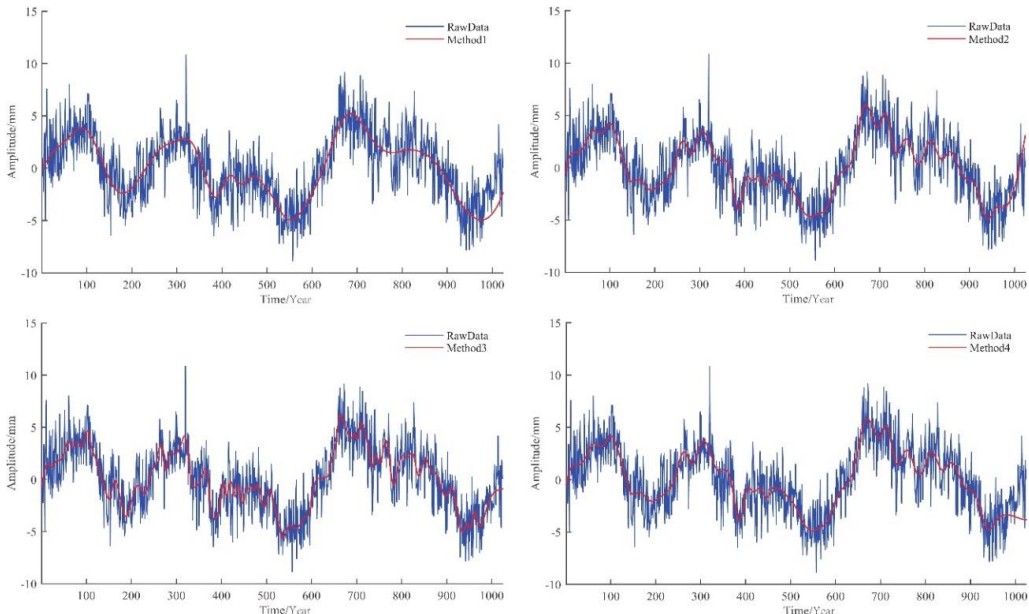

**Figure 13.** Signal sequence graph of simulation data I after noise reduction in different methods.

In summary, experiments with three sets of simulation data demonstrated the ability of the three new algorithms proposed and contained in GNSS-TS-NRS to reduce signal noise and enhance the practicability and operability of GNSS-TS-NRS. On the other hand, the three accuracy indicators of correlation coefficients, RMSE, and SNR quantitatively showed that the three improved algorithms outperformed the traditional EMD method. The software also drew the graphics of the signal sequences processed by different methods to intuitively reflect the noise reduction effect.

*4.2. Test GNSS-TS-NRS with Real GNSS Data*

In order to verify the practicability and reliability of the software, experiments were conducted on the GNSS-TS-NRS platform using the data of the BJFS station from March 1999 to March 2006, which spanned 7 years. Before the noise reduction analysis, in order to reduce the negative impact of the gross error on the experiment, the gross error elimination tool in GNSS-TS-NRS was used to eliminate the gross error of the test data. The discrimination criteria included four algorithms: 3 Sigma, 5 Sigma, 3 IQR, and MAD. After the gross error was eliminated, the software generated a report of the gross error elimination rate (Table 3) and marked the position of the gross error in the entire time series (Figure 14). Compared with the EN components, the U component shows relatively more gross errors, and the E and N direction sequences did not detect gross error points. We chose the U direction data processed by the 3 Sigma rule as the experimental data.

**Table 3.** Gross error rate of BJFS station under different algorithms.

| Criterion | 3 IQR | 3 Sigma | 5 Sigma | MAD |
|---|---|---|---|---|
| Error rate (%) | 0.196 | 0.352 | 0.196 | 0.274 |

We imported the above data into GNSS-TS-NRS and analyzed the noise reduction with four algorithms to compare the noise reduction effects of different methods. First, the correlation coefficient between the *IMFs* obtained by the EMD method and the original sequence is shown in Figure 15. The correlation coefficient took the minimum value for the first time when the *IMF* was 4. Therefore, according to the correlation coefficient criterion, K = 4 was judged to be the boundary *IMF* value. The first to fourth *IMF* were recognized as high-frequency noise components, so they were removed,

and the next 7 *IMFs* components were superimposed to form a noise-reduced signal. Figure 16 shows the *IMFs* of raw data through EMD.

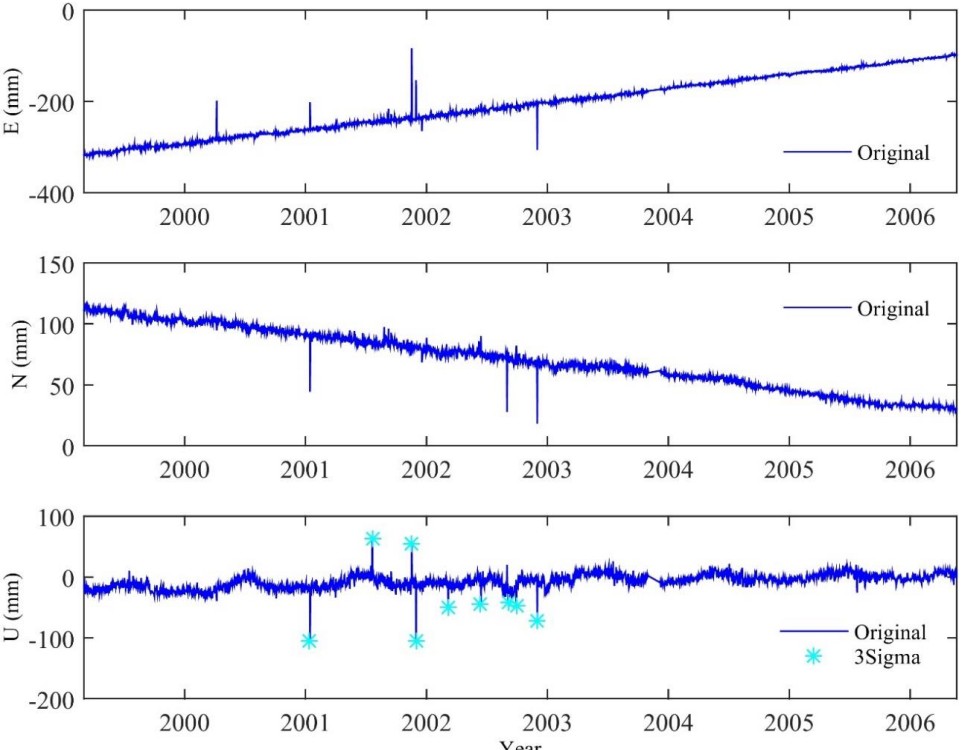

**Figure 14.** The BJFS station coordinate time series is judged by the 3 IQR rule for gross errors, and blue stars indicate gross errors.

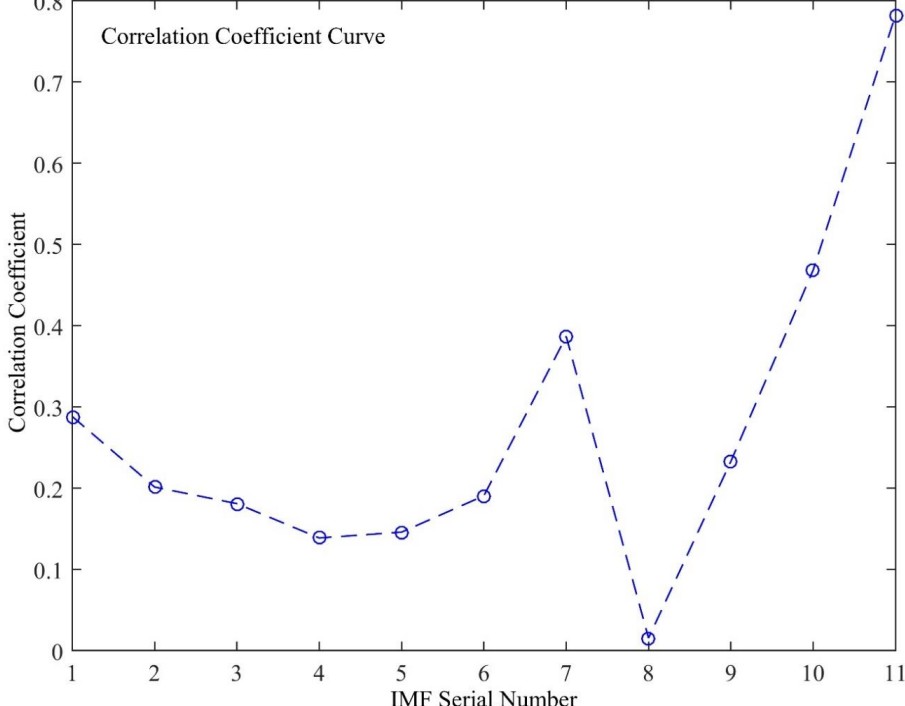

**Figure 15.** The correlation coefficient between the *IMF* obtained by empirical mode decomposition (EMD) and the original signal.

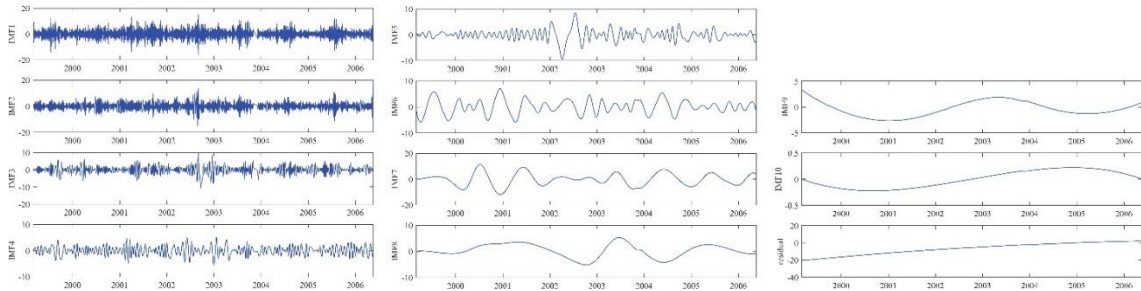

**Figure 16.** The *IMFs* of BJFS station in U direction from EMD.

Method 2 iterated on the basis of the first EMD decomposition, superimposed the second to the boundary *IMF* component, and continued to decompose until the correlation coefficient showed a monotonic trend or the number of *IMFs* decomposed was less than 3, and then the iteration was stopped. In this signal, decompositions were carried out 14 times. In each decomposition, we attained a set of *IMFs* from high frequency to low frequency. The boundary *IMF* value was calculated by a correlation coefficient rule. Last, the set of 14 low-frequency *IMFs* added to the noise reduction signal. In Method 3, the software separately calculated the average period, energy density, and their product. Then the method calculated its threshold. The results showed that: k = 3 was the demarcation point, that is, the boundary *IMF* K = 4, which was similar to the result of Method 1. Method 4 attained different results, and judged the boundary *IMF* K = 3. After completing the calculations, the software automatically drew the original signal sequence and the signal sequence obtained by different methods for visual comparison as shown in Figure 17.

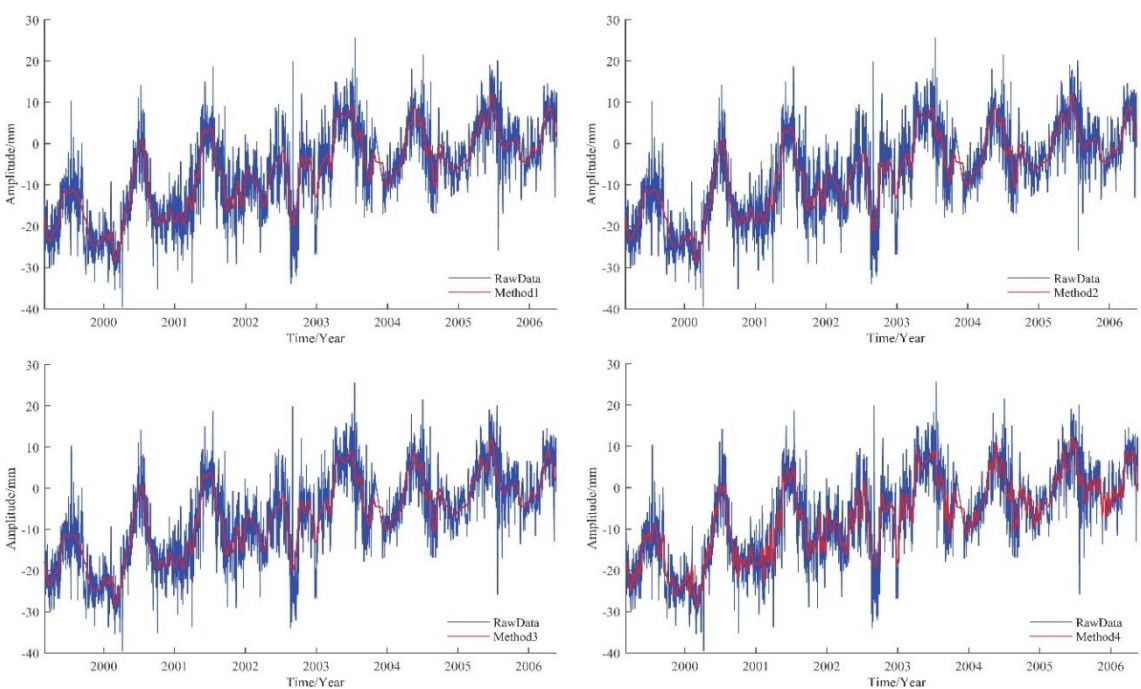

**Figure 17.** BJFS station U direction time series processed by four noise reduction methods.

As mentioned above, the boundary *IMF* values obtained by Methods 1 and 3 were equal, so the graphs of Methods 1 and 3 overlap. From Figure 17, we can see that different the noise reduction methods were obviously efficient in noise reduction for the time series of the U direction of the BJFS station. The graph is clearly gentler, and more intuitively reflects the changing trend of the station. The results of Methods 2 and 3 are basically similar to Method 1, which proves its noise reduction effect. Method 4 obviously reflects more detailed changes and achieves a better noise reduction effect.

Besides the qualitative analysis of graphic comparison, we quantitatively analyzed the noise reduction effect with three indicators: Correlation coefficient, root mean square error, and signal-to-noise ratio. In Table 4, compared with Methods 1 and 3, Method 2 obtained a lower RMSE and a higher SNR, indicating that the overall signal was more stable, although the correlation coefficients were basically the same. Similarly, Method 4 also obtained a performance gain, with the correlation coefficient increasing by 0.0091, RMSE decreasing by 0.2723, and the SNR increasing by 0.7037, indicating that the selection of boundary *IMF* was more accurate. In summary, the reliability of the three improved EMD-based noise reduction methods proposed in this paper were verified.

**Table 4.** Accuracy evaluation results of measured data.

| Method | Evaluation Indicator | | |
|:---:|:---:|:---:|:---:|
| | $\rho_k$ | RMSE | SNR |
| 1 | 0.9149 | 5.1463 | 5.0256 |
| 2 | 0.9153 | 5.1327 | 5.0839 |
| 3 | 0.9149 | 5.1463 | 5.0256 |
| 4 | 0.9240 | 4.8740 | 5.7292 |

## 5. Conclusions and Future Research Direction

We designed an open-source MATLAB-based GNSS-TS-NRS software, and the software showed good interactivity. With batch processing for GNSS time series noise reduction and analysis, it could also be extended to work with other geodetic time series. For GNSS-TS-NRS, we implemented classic EMD and EEMD algorithms, and we also proposed three improved algorithms based on EMD and acquired good results. In addition, we realized 5 spatial filtering analysis methods with MATLAB, i.e., the stacking filtering method, weighted stacking filtering method, correlation weighted stacking filtering method, distance weighted filtering method, and principal component analysis. Last, GNSS-TS-NRS software provided us with several time series processing tools, and it provides useful tools for new users to explore GNSS time series noise reduction and application.

We intend to share the GNSS-TS-NRS software with the scientific community to introduce new users to the GNSS time series signal processing and noise reduction technique. The software can also be extended to work with other geodetic time series, for example, tide gauges observations for sea level rise study, illustrating that users can easily adapt the software for other purposes. The software is available at https://github.com/CL-Xiong/GNSS-TS-NRS. Apart from the improvements from the current modules, some new time series analysis and prediction methods, e.g., independent component analysis, wavelet decomposition, deep learning and machine learning, and the prophet model will be integrated into GNSS-TS-NRS in the future. In addition, a future version of the software will be available in Python together with a new application for sea level rise study. We hope that the GNSS-TS-NRS software can benefit the users who want to perform post-analysis of geodetic time series, and we have thus developed it under an open-source framework.

**Author Contributions:** X.H., J.-P.M., and C.X.: Methodology, software design, and writing–original draft preparation; K.Y., T.L., and S.Z.: review and editing the manuscript. X.M., F.M., and H.C.: provision of study materials, computing resources, and data analysis tools. All authors have read and agreed to the published version of the manuscript.

**Funding:** This work was sponsored by the National Key R&D Program of China (2018YFC1503600), National Science Innovation Group Foundation of China (41721003), National Natural Science Foundation of China (42061077, 42064001, 42074006, 41904171, 41804007), Natural Science Foundation of Jiangxi Province (20202BAB214029, 20202BABL211007), and the Key Laboratory of Geospace Environment and Geodesy, Ministry of Education, Wuhan University (Grant No. 18-02-05).

**Conflicts of Interest:** The authors declare no conflict of interest.

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
