# Peer review of "GNSS-TS-NRS: An Open-Source MATLAB-Based GNSS Time Series Noise Reduction Software"

_remotesensing, doi:10.3390/rs12213532_

Round 1

Reviewer 1 Report

Dear Editor, my comments for Authors:

  1. Abstract must include the information about obtained results from paper !
  2. All acronyms must be explained in text, e.g. GNSS, EMD, EEMD, etc.
  3. Introduction must include the information about novelty of paper ! For example new solution, new strategy, new algorithms etc. It must be underline in Introduction part.
  4. Equation (4), what means xk, yk, zk ?
  5. Equations (6-9), all symbols must be explained in the text, e.g. what means "K"?
  6. Equations (11-13), all symbols must be explained in the text, e.g. what means "t"?
  7. Figures 3 and 4 is illegible.
  8. Figure 5 is illegible.
  9. Figure 6 is illegible.
  10. Figure 7 is illegible.
  11. Figure 8 is illegible.
  12. Please better descibe the results into Figures 3-8.
  13. Equation (27), please explain all symbols.
  14. Figure 13 is illegible.
  15. Figure 16 is illegible.
  16. Figure 17 is illegible.
  17. Chapter "Discussion and Concluding remarks" must be corrected. I don't see the colnclusion. What is a novelty of paper. Please compared obtained results from another solution from another software, if it possible. I want to know how your solution improved the noise reduction in GNSS observations.
  18. Please check the numbering of Chapter, e.g. "4.Noise Reduction Test and Result", "4. Discussion and Concluding remarks".

Currently, the Authors must revised this paper.

Author Response

Thanks for your comments, please see the attached R1 document.

Reviewer 2 Report

This paper introduces us an open source matlab code for GNSS time series noise reduction. However, until now, the matlab code hasn’t be uploaded yet, in the GitHub. This is a major disadvantage of the paper, especially as paper’s presented idea is consentrated more on describing the implemented software. Furthermore, the paper is written as a technical manual of the GNSS-TS-NRS software and not as an academic paper. The authors should consider re-organizing their paper and provide an application scenario with which they could evaluate their software and in addition compare the results with already existing software/approaches. Overall, the comments are summarized below:

  1. In the introductory section the authors should consider emphasizing the contribution of their work and mention the comparative advantage of their method. The advantages should not be limited only in details related to the interface but also be targeted to the methodology approach used in GNSS-TS-NRS in contrast to other already existing approaches in the field of GNSS timeseries analysis.
  2. The related work is limited, thus the authors should include additional researches dealing with GUIs and services creation for GNSS timeseries modeling and visualization. Examples are bellow:
  • Lanagran-Soler, F., Vazquez, R. and Arahal, M.R., 2015. A Matlab Educational GUI for Analysis of GNSS Coverage and Precision. IFAC-PapersOnLine, 48(29), pp.93-98.
  • Kaselimi, M., Doulamis, N., Delikaraoglou, D. and Protopapadakis, E., 2018. GNSSGET and GNSSPLOT Platforms-Matlab GUIs for Retrieving GNSS Products and Visualizing GNSS Solutions. In VISIGRAPP (5: VISAPP) (pp. 626-633).
  • Dabove, P., Di Pietra, V. and Piras, M., 2020. GNSS Positioning Using Mobile Devices with the Android Operating System. ISPRS International Journal of Geo-Information, 9(4), p.220.
  1. The authors should consider adopting the structure used in scientific journals. The section experimental results are absent. The authors should consider providing a case study and an experimental set up in which they will evaluate their software and to compare their results with the already existing software/approaches, under real data and real circumstances/scenarios.

Author Response

Thanks for your helpful comments, we have improved the manuscript according too your comments, please see the attached R2 document.

Reviewer 3 Report

This paper has a potential to be accepted, but some important points have to be clarified or fixed before we can proceed and a positive action can be taken.

  1. The GNSS-TS-NRS software is not found athttps://github.com/CL-Xiong/GNSS-TS-NRS.
  2. The authors seem to disregard or neglect some important results that have been recently achieved in this specific field. For example authors should not ignore the HECTOR software (http://segal.ubi.pt/hector/). We need to understand how the proposed approach is related with this result.
  3. It is really unclear to me: "we developed three improved algorithms based on EMD". This is an important point.
  4. Does the user need to know what type of temporal correlated noise exists in the observations.
  5. Have you compared the uncertainty of the precision before and after the noise reduction.
  6. Page 12 <4.Noise Reduction Test and Result> and page 18 < Discussion and Concluding remarks> ?
  7. I think that if the authors wish this paper is well considered by experts in the GNSS time series field, more attention should be devoted to discuss the application real The provided simulative results are not completely convincing to me. Again they are too vague and generic.

Provided the above questions are answered and problems are fixed, the paper can be reconsidered for publication.

Author Response

Thanks for your  comments, we have modify the manuscript  according too your comments, please see the attached R3 document.

Reviewer 4 Report

The manuscript presents a tool for handling GNSS time series. As such, the methods are adequate and the process clear, but there are some things that make the text hard to decipher and thus, calls for some revision.

First of all, I would like to argue that MATLAB is a commercial software that comes with license payments. Thus, the reasoning and statement on page 2 (lines 62) on how this package is independent of commercial software is misleading. MATLAB is widely used and available, through different institutions etc, but it is not freely available. Using MATLAB is fine, just rephrase the reasoning. I would have also liked to see references to Hector, CATS, etc programs in the first sentence instead of GGMatlab, especially when the references point to other programs.

Chapter 2 is a bit strange. I would prefer seeing the installation instructions and suggestions in the correct program version to be in the readme-file of the program, not in the article. There was also lot of advertising-like text in the chapter.

In chapter 2 are five modules listed, but the subchapters of chapter 3 don't agree with them at all. In chapter 2 you talk about modules and subchapters are "this and that" model? Is that a typo? What is c-located? Co-located maybe? Is the common mode error mitigation module explained in 3.1? Check the names, please, to make the text readable and comprehendable.

What are xk, yk and zk  in formula (5)?

Lots of abbreviations are explained too late: EMD, EEMD, TS, IMF...

Page 6, line 210 says "return to step 2" Does that mean line 204? Please add step numbers, if you are going to refer to them.

Formula (11) has T as well as(23) and (24), I assume they are not the same T?

The discussion was very short, but since there really were no results as such, it is adequate.

Author Response

Thanks for your helpful comments, we have corrected the manuscript according too your comments, please see the attached R4 document.

Round 2

Reviewer 1 Report

I accept the paper in current form. 

Author Response

we make the relate correction, and attached the file 'Reply to Academic Editor Notes'

Reviewer 2 Report

The authors have addressed all my previous comments and they have significantly improved the content of the paper, thus I recommend the paper to be approved on its current form.

Author Response

(The authors gave the same response as above.)
